# Biocompatible and Biodegradable Magnesium Oxide Nanoparticles with In Vitro Photostable Near-Infrared Emission: Short-Term Fluorescent Markers

**DOI:** 10.3390/nano9101360

**Published:** 2019-09-23

**Authors:** Asma Khalid, Romina Norello, Amanda N. Abraham, Jean-Philippe Tetienne, Timothy J. Karle, Edward W. C. Lui, Kenong Xia, Phong A. Tran, Andrea J. O’Connor, Bruce G. Mann, Richard de Boer, Yanling He, Alan Man Ching Ng, Aleksandra B. Djurisic, Ravi Shukla, Snjezana Tomljenovic-Hanic

**Affiliations:** 1School of Physics, University of Melbourne, Parkville, VIC 3010, Australia; r.norello22@gmail.com (R.N.); jptetienne@gmail.com (J.-P.T.); tkarle@unimelb.edu.au (T.J.K.); 2School of Science, Engineering and Health, RMIT University, Melbourne, VIC 3000, Australia; amanda.abraham@rmit.edu.au (A.N.A.); ravi.shukla@rmit.edu.au (R.S.); 3Department of Mechanical Engineering, University of Melbourne, Parkville, VIC 3010, Australia; ewlui@unimelb.edu.au (E.W.C.L.); k.xia@unimelb.edu.au (K.X.); 4Faculty of Science and Engineering, Queensland University of Technology, Brisbane, QLD 4000, Australia; phong.tran@qut.edu.au; 5Department of Chemical and Biomolecular Engineering, Particulate Fluids Processing Centre, Melbourne, VIC 3010, Australia; a.oconnor@unimelb.edu.au; 6The Department of Surgery, University of Melbourne, Parkville, VIC 3010, Australia; bruce.mann@mh.org.au; 7The Breast Service, Victorian Comprehensive Cancer Centre, Parkville, VIC 3052, Australia; Richard.DeBoer@wh.org.au; 8Department of Physics, Southern University of Science and Technology (SUSTech), Shenzhen 518055, China; heyl3@mail.sustc.edu.cn (Y.H.); ngamc@sustc.edu.cn (A.M.C.N.); 9Department of Physics, University of Hong Kong, Pokfulam Road, Hong Kong, China; dalek@hku.hk

**Keywords:** magnesium oxide, fluorescence nanoparticles, bioimaging, biomarking, confocal microscopy, cancer cells

## Abstract

Imaging of biological matter by using fluorescent nanoparticles (NPs) is becoming a widespread method for in vitro imaging. However, currently there is no fluorescent NP that satisfies all necessary criteria for short-term in vivo imaging: biocompatibility, biodegradability, photostability, suitable wavelengths of absorbance and fluorescence that differ from tissue auto-fluorescence, and near infrared (NIR) emission. In this paper, we report on the photoluminescent properties of magnesium oxide (MgO) NPs that meet all these criteria. The optical defects, attributed to vanadium and chromium ion substitutional defects, emitting in the NIR, are observed at room temperature in NPs of commercial and in-house ball-milled MgO nanoparticles, respectively. As such, the NPs have been successfully integrated into cultured cells and photostable bright in vitro emission from NPs was recorded and analyzed. We expect that numerous biotechnological and medical applications will emerge as this nanomaterial satisfies all criteria for short-term in vivo imaging.

## 1. Introduction

Nanoparticles (NPs) have attracted considerable interest as diagnostic and therapeutic tools for biomedicine [1,2,3]. Using the fluorescence from NPs as an in vitro and in vivo marker is an important tool in the determination of the behaviour and function of cells in their natural environment. Laser-induced fluorescence, a spectroscopic method that uses optical emission excited by absorption of laser light, is one of the key avenues to understand cellular processes [3]. However, in a biological cell the presence of components such as collagens produce fluorescent background signals. These components typically absorb light in the range 300–500 nm and fluoresce at 400–550 nm [4]. Therefore, it is essential for the probe to absorb the light at wavelengths longer than 500 nm and to emit light at wavelengths longer than 600 nm. Additionally, for in vivo applications the light emission should be in near-infrared range (NIR), from 700 to 900 nm, as NIR light penetrates centimetres into tissue, whereas visible light can only travel microns [5].

Organic dyes meet some requirements but they suffer from photobleaching [6]. Quantum dots (QDs) have been investigated due to their brightness but the major obstacle in their clinical use remains their toxicity [7]. Optical centres in nanodiamond are being increasingly used for a variety of advanced biotechnology due to their photostability, biocompatibility and body temperature emission [8], but the nanodiamonds do not biodegrade.

Recent research has begun to use metal oxides for this purpose, due to their biocompatibility, biodegradability, therapeutic properties and desirable optical properties [9]. The therapeutic properties of metal oxides have been well explored for many industrial and biomedical applications [10]. However, their intrinsic fluorescence is unstable [11]. The primary challenge involves engineering the properties of the fluorescent emitters in metal oxides to obtain photostable body temperature emission [11,12]. It has been demonstrated that several metal oxide dopants can be successfully integrated in one compound to achieve photostability. Biocompatible and stable, in phosphate buffered saline, ZnO/MgO/Fe_2_O_3_ composite nanocrystals showed not only stable luminescence but also magnetic properties confined to a single NP at room temperature [13]. Another approach to obtain stable luminescence is coating of NPs with a biocompatible polymer that ensures not only their stabilization in solution but is also expected to enhance their optical emission properties [14]. However, full and long term encapsulation is not suitable if the advantage of biodegradability and/or therapeutic properties of these NPs are to be used in biomedical applications. Recently, photostable in vitro single-photon emission has been observed from optical defects within ZnO NPs derived from ion implantation followed by thermal oxidation [15]. Even though, stable fluorescence of these defects has great potential for various applications, for their practical realization this material needs to be widely available.

Magnesium oxide (MgO) is also known to be biodegradable [16], has a long fluorescence decay time (in the order of milliseconds) [17] and is widely available. Currently, the only research conducted into the use of MgO in biological fields has been centred on its therapeutic antibacterial [17], cytotoxic [18] and antithrombotic properties [19]. Biodegradable and low-toxicity MgO NPs express highly desirable properties for cancer labelling [18,20,21]. MgO NPs have been used in proof-of-principle nano-cryosurgery for cancer treatment [20] and ZnO/MgO composite NPs have also been used for cancer labelling [18].

Extensive defect photoluminescence studies on different forms of MgO have been performed since the late 1960s. These include studies on single crystal [22], nanocrystalline [23] and thin film [23]. MgO synthesised by a multitude of methods. Among these studies, the most prominently discussed point defects are the anionic F and F^+^ centres, which contain an oxygen vacancy. Though these two defects are the most optically active and researched, their short excitation wavelengths are not suitable for in vivo imaging [5].

However, certain optical defects in MgO, are known to satisfy many of the requirements of a fluorescent in vivo marker, including an absorption spectrum up to 550 nm, and spectrally resolvable emission lines above 700 nm. Of importance to the research presented in this manuscript are much less studied defects, the vanadium (V^2+^) and chromium (Cr^3+^) ion substitutional defects. To the author’s knowledge only one research team has studied the near-infrared signal from a bulk MgO sample excited by 325 nm and 532 nm radiation [22]. This study assigns the 688 nm and 700 nm lines, only excited by 532 nm laser, and the broad 800 nm line to Cr^3+^ substitutional defects. Additionally, the lines above ~850 nm have been assigned to V2+ substitutional defects. Emission lines around 700 nm have previously been discussed, though in this case they were only excitable at low temperatures, 5–7 K [24]. They were attributed solely to Cr^3+^ defects.

MgO NPs have a unique combination of biocompatibility and biodegradability in addition to being intrinsically fluorescent. The fluorescence provides the ability to track them in biological cells and tissues. In this manuscript we describe the optical characterisation of ball-milled and commercially available MgO NPs, and their suitability for bioimaging applications. We report effective cell internalization of MgO NPs in both healthy cells and cancer cells and demonstrate their imaging capability at cellular level.

This opens the way for several future applications of MgO NPs. One of the potential applications of these biocompatible and biodegradable NPs is for cutting-edge fluorescence guided surgery. The pressing need for the fluorescence guided surgery through the use of fluorescent nanoparticles has been clearly identified for two main reasons: (a) for tumours that are difficult to differentiate from adjacent normal tissues, such as for breast cancer, and (b) for tumours that are close to crucial structures, such as for brain tumours [5]. In such applications, it is envisioned that these fluorescent, non-toxic, degradable MgO NPs could be manufactured to specifically target the tumor through size-dependent enhanced permeability and retention (EPR) effects and/or binding of ligands that recognize receptors on cancer cells. These NPs could also be engineered to carry chemotherapy drugs that are thus delivered locally to the targeted tumour tissue to improve treatment efficacy and minimize systemic toxicity.

## 2. Materials and Methods

### 2.1. Preparation of MgO Nanoparticles

Ball-milled NPs were produced from single MgO (100) crystals, purchased from MTI Corporation (Richmond, CA, USA), with a size of 10 mm × 5 mm × 0.5 mm. MgO crystal was grown by arc melting method and nature cooling. The single crystals were broken into approximately 2–3 mm sized pieces by tapping with a metal hammer to allow for easier ball milling. Ball milling was carried out under vacuum using a high-energy planetary mill with tungsten carbide jars and balls (6 mm diameter, ball-to-weight ratio of 20:1) for 2 h at 400 rpm. After milling, ~0.03 g of MgO powder dispersed in 8 mL of isopropyl alcohol (IPA) was centrifuged to determine the range of particle sizes. Centrifugation (Beckman Coulter, Indianapolis, IN, USA) was performed at a range of speeds (1000–4000 rpm) to separate the larger particles from the smaller ones. Both the pellet (the larger particles which made their way to the bottom of the tube) and the supernatant (the smaller particles left in IPA) sizes were measured using Dynamic Light Scattering (DLS) measurement (Zetasizer, Malvern Scientific, Sydney, Australia).

Commercial MgO NP samples were purchased from MTI Corporation with an average size of 30 nm (20–40 nm). The NP powder was dispersed into water and sonicated for 5 min to make a homogenous solution with a final concentration of 2 mg/mL.

### 2.2. Cell Culture

Human keratinocyte (HaCaT) cells and prostate cancer (PC-3) cells, in their active phase of growth, were maintained in Roswell Park Memorial Institute 1640 (RPMI 1640) media and 3T3 fibroblast cells were cultured in Dulbecco’s Modified Eagle Medium (DMEM) (Carlsbad, CA, USA). In both cases, media was supplemented with 10% foetal bovine serum (Invitrogen, Carlsbad, CA, USA) and 1% penicillin-streptomycin (Life Technologies, Carlsbad, CA, USA) and cells were maintained at 37 °C, in the presence of 5% CO_2_ and 95% humidified air. For imaging, cells were treated with 1 mg/mL of the MgO NPs for 12 h, after which, cells were washed in in phosphate buffered saline (PBS) followed by imaging of live cells.

### 2.3. Confocal Imaging

Confocal fluorescence scanning was performed using a customized confocal microscope. A 532 nm continuous wave frequency doubled Nd:YAG laser was used for sample illumination at a power of 200 μW, through a 100×, 0.95 NA objective. The samples on silicon (Si) and cover glass substrates were mounted on a computer-controlled stage facing the objective perpendicularly and equipped with an xyz closed loop positioner with 100 μm travel in each direction and step size resolution of 100 nm. The in-plane optical resolution was approximately 300 nm. The fluorescence from the samples was collected with a Single Photon Avalanche Photodiode (SPAD) in the red to near infrared wavelength range. SPADs were used to measure the fluorescence intensity in units of photons detected per second. Avalanche photodiodes are used for single photon counting with dark count rates well below 1 k counts/s. The intensity traces have been optimized (at a range of power values from 10 µW to 1 mW) with reference nanodiamond samples with single photon emitting color centres. The emission was filtered by a 560 nm long pass filter to remove the incident beam before being detected.

### 2.4. Wide Field Fluorescence Imaging

Widefield imaging was performed with a commercial fluorescence microscope (Eclipse Ti-U, Nikon, NY, USA), using a 532 nm Verdi laser operating with a typical power density of 45 µW/mm^2^ in a temperature controlled environment of 37 °C. A 5× beam expander was used to expand the excitation beam to 10 mm diameter before focusing the excitation light onto the back aperture of the 100×, 1.45 NA (Nikon) oil immersion objective through a dichroic mirror (Semrock-Di02-R561–25×36, New York, NY, USA). A focusing lens (f = 300 mm) focused the excitation light onto the back aperture of the objective creating a uniform widefield illumination. The laser power P = 300 mW was spread over 80 × 80 μm^2^ area and the widefield fluorescence image was detected with an sCMOS camera (Neo, Andor, Abingdon, UK). The setup is also equipped with a white light source that was first used to find a particular area of interest on the sample. The laser was then turned on to image the fluorescence scan for the particular area. Fluorescence imaging of the NPs was performed several microns above the surface of the coverglass where the fluorescent NPs came into focus. Care was taken to ensure the focal point did not exceed the average height of the cultured cells to make sure that only the NPs inside the cells are being imaged.

### 2.5. Lifetime Measurements

Fluorescence lifetimes were measured in a confocal microscope similar to that described in Section 2.3, adapted to include a pulsed laser source and a time resolved photon counting module (MCS6A, FAST ComTec, München, Germany). The pulsed green excitation source was constructed from a 520 nm pigtailed laser diode (Thorlabs LP520-SF15) and a Bias-T circuit (LDM9LP, Thorlabs, Newton, NJ, USA) that allows fast current modulation. A pulse generator (PulseBlaster ESR-PRO 500 MHz, SpinCore, Gainesville, FL, USA) was used to modulate the diode current and create 50-ns laser pulses with a measured fall time of ≈0.5 ns and a repetition rate of 2 MHz. Fluorescence decay traces were obtained by time-tagging the photon arrivals following the end of each pulse and averaging the signal over a total integration time of about 10 s. The decay traces were then fitted to a single exponential decay to extract the fluorescence lifetime, or, when the single exponential fit was not satisfactory, to a biexponential decay model. About 20–30 NPs were measured for each type (commercial and ball-milled).

### 2.6. Absorption and Photoluminescence Spectra

Absorption spectral analysis: The absorbance spectra of the two types of MgO NPs were measured using a UV–visible spectrophotometer (Varian Cary 50MPR, Agilent Technologies, Santa Clara, CA, USA).

Photoluminescence analysis: The photoluminescence signal from the confocal microscope was coupled into a spectrometer and detected by a CCD camera (SpectraPro 2300i with Pixis100 Camera, Acton, NJ, USA).

### 2.7. Stability in Cell Culture Media and Cytotoxicity Analysis in Cancer Cells

To study the stability of the MgO NPs under cell culture conditions, the MgO NPs were incubated in PBS and PBS containing 10% foetal bovine serum at 37 °C. Dynamic light scattering (DLS) and zeta potential measurements were conducted after 24 h incubation, on a Malvern Zetasizer. The measurements were performed at 25 °C and each sample and 6 scans were performed for each sample.

The PC-3 cells were seeded into a 96 well plate at a seeding density of 1 × 10^4^ cells/well. The cells were allowed to attach for 24 h. The cells were then treated with the increasing concentrations of MgO NPs (0–1000 μg/mL) in growth media for 24 h. After incubation, MTT (3-(4,5-dimethylthiazol-2-yl)-2,5-diphenyltetrazolium bromide) assay was used to determine the toxicity. Briefly, 10 μL of MTT solution (Life Technologies, 5 mg/mL in PBS) was added to each well and incubated at 37 °C for 4 h. The media was removed and the purple formazan crystals thus formed, were dissolved in 100 μL of acidified isopropanol. The absorbance was recorded at 570 nm and results were analysed by plotting a histogram of concentration verses relative decrease in the absorbance values at 570 nm.

### 2.8. Cellular Staining

Cells were grown in a Lab-Tek Chamber Slide system (Nunc, MA, USA) at a seeding density of 1 × 10^5^ per well for 24 h. Human cancer cell line was PC-3 cells and the normal cells were mouse cell line J774. For staining, the cells were fixed in 4% paraformaldehyde for 10 min and washed thrice in PBS. Cells were permeabilised with 0.02% Triton X-100 in PBS for 90 s, followed by blocking with 2% BSA in PBS for 30 min. Cells were incubated with E-cadherin rabbit antihuman monoclonal antibody (1:50 dilution) (Abcam, Cambridge, UK) for 30 min and washed twice with PBS. Secondary antibody was goat antirabbit Alexafluor 594 (1:500 dilution) (Life Technologies, Carlsbad, CA, USA) for 1 h. The normal cells were unstained.

## 3. Results

### 3.1. Structural Characterization

Commercial and the ball-milled MgO NPs are imaged using SEM as shown in Figure 1. High resolution SEM images are obtained for individual MgO nanoparticles. 

An accelerating voltage of 500 V is used to collect images at high magnifications of 100,000× to 350,000×. For commercial MgO, spherical nanoparticle shape is observed with a size distribution of 30 ± 3 nm to 140 ± 10 nm for the individual NPs, as can be seen by in Figure 1a. For ball-milled MgO, irregular shaped crystals are observed with a relatively broader size distribution ranging from 70 ± 10 nm to 230 ± 30 nm, as shown in Figure 1b.

Additional TEM images are provided in Appendix A. In all samples, variation of shapes and sizes can be observed. This is common not only for a high energy ball milling, but also for many methods of nanoparticle synthesis. Additionally, high energy ball milling has been shown to dramatically change the chemical and physical characteristics of the metal oxide [25]. As confirmed with Dynamic Light Scattering (DLS) measurements, size distribution varies from 760 nm to 1640 nm for the pellet samples, and 100 nm to 300 nm for the supernatant samples with centrifuge speeds between 1000–3000 rpm. DLS results are shown in Table 1.

There is a general decrease in particle size as the centrifuge speed increases, implying the larger particles are being eliminated with each increase in speed. As expected, the highest centrifuge rate supernatant has the smallest particle size. Polydispersity index (PDI) for supernatant appears to be much smaller (<0.2) than pallet samples, because the former are monodispersed smaller sized particles.

The DLS measurements for the commercial MgO powder yielded a size distribution averaged around 1.0 ± 0.5 µm. It is very common for commercial nano powders to have a large size variation consistent with the fact that many low cost synthesis methods suitable for mass production do not result in mono disperse nanoparticles [26,27,28].

An average zeta potential value of 13.8 ± 0.2 for the commercial MgO and 15.83 ± 4.27 mV for the ball-milled MgO NPs is observed in water, which is expected to contribute to partial stabilization of the NPs in the solution by maintaining the electrostatic repulsion between the particles.

Arc melting is known to introduce impurities of calcium, aluminium, silicon, vanadium and chromium. The inductively coupled plasma mass spectrometry (ICP-MS) was conducted on the ball-milled MgO nanopowder. These results are presented in Table 2 where each possible impurity is recorded as a percentage present within the sample. Most importantly, the results show the presence of chromium and vanadium impurities within the sample.

### 3.2. Confocal Microscopy

To investigate the fluorescence properties of MgO, both commercial and ball-milled NP samples were deposited (drop-cast) on the Si substrate and were scanned with a confocal microscope. Figure 2a,b show the 100 × 100 μm^2^ confocal scans of the commercial and ball-milled samples respectively. Both images display bright fluorescent NPs with a maximum of the order of 10^6^ counts/s for the ball-milled, and commercial MgO sample. Light blue background in both figures indicates low back-ground counts of the order of a few 10^3^ counts/s.

To examine the detailed fluorescence characteristics of the samples, zoomed images were recorded for both samples. The insets of Figure 2a,b show 10 × 10 μm^2^ regions on the commercial and ball-milled samples. The bright circular shaped NPs were seen in the magnified confocal scans of both samples. The individual, non-agglomerated NPs provided photostable bright counts as shown by the counts trace of Figure 3a,b.

Commercial MgO NPs were brightly fluorescent with emission counts of the order of a few million counts per second. The ball-milled MgO NPs exhibited a range of emission counts from 50 k–250 k counts per second. For both types of samples, 25–30% of the observed NPs exhibited photo-bleaching behavior, however majority of the NPs were found to be photostable.

The emission counts are a function of excitation power and are expected to increase with increase in excitation power. However, we were limited to use lower excitation power in order to ensure noninvasive imaging consistent with our in vitro cell assay. Moreover, the NPs are fluorescent due to the presence of emission centres in their crystal lattice. The emission counts per second emitted by a NP is dependent mainly on the size of the NP, the number of fluorescent emitters present inside that NP, and the location of emitter inside the NP [29,30].

### 3.3. Spectral Analysis

#### 3.3.1. Absorbance Spectra

The characteristic absorbance and emission spectra were recorded for MgO NPs. The measured absorbance of MgO NPs in the visible to infrared range are shown in Figure 4.

It can be seen that the recorded absorbance for the NPs increases at shorter wavelengths, possibly due to increasing scattering at these wavelengths, as expected for particles much smaller than the wavelength of light used [31]. However, both commercial and ball-milled NPs do exhibit absorbance in the visible to infrared range. This implies that the NPs fulfil the requirement as probes capable of absorbing light at wavelengths longer than 500 nm, which is essential for low background bioimaging of cells and tissues.

#### 3.3.2. Emission Spectra

The photoluminescence (PL) spectra were recorded for both commercially available and ball-milled MgO NPs, as presented by the plots of Figure 5a,b, respectively. The spectral characteristics were investigated to analyze any differences in the emission properties between these two samples. Both samples exhibited broad fluorescence between 650–900 nm. However the PL spectrum of commercial MgO NPs revealed lines above 800 nm while ball-milled MgO revealed lines at 688 nm and 700 nm. The zero phonon line (ZPL) assigned as the magnetic—dipole no-phonon 2E-4A2 transition with the lines at 688 nm and 700 nm have been attributed to Cr^3+^ substitutional defects in cubic (R-line) and non-cubic (N-line), respectively; observed only during the 532 nm excitation [23]. These lines are followed by broad fluorescence with a maximum around 800 nm that is reported to be independent of the excitation wavelength [23]. The MgO (100) crystal, grown by arc melting, is known to contain chromium impurities [22]. 

Lines above 800 nm, displayed for the commercial nanoparticles, have been attributed to the presence of V^2+^ substitutional defects [22]. According to literature, the intensity of the sideband lines at 837 nm and 852 nm are most likely the result of a phonon assisted transition. The line at 870 nm has been attributed to a magnetic dipole allowed and described as the ZPL of the 2Eg-4A2g transition in the MgO:V^2+^ system (S line). The lines at 891 nm and 908 nm have been reported to be the sidebands of the R line and assigned to an electric dipole allowed transition [22]. 

Hence, the results yielded the presence of two different emission centre types in the commercially available and ball-milled samples. Most importantly, both samples showed PL above 700 nm which is highly suitable for in vitro and in particular in vivo imaging applications. Moreover, using combinations of different colour centres enables multicolour experiments often needed for molecular targeting or imaging of specific cells.

Some reports underline the difficulties of maintaining desired nanomaterial properties during handling and processing due to their dynamic nature [32]. In particular, blue shift in PL maximum was reported for l MgO cubes (size dependant).

The fluorescence quantum yield or efficiency (QE) of the MgO NPs is experimentally calculated. The QE of the MgO NPs was measured by comparing their photoluminescence spectra to those of nitrogen vacancy (NV) centre in nanodiamonds. The NV centre in nanodiamonds [33] have a known quantum yield of 0.7 [33]. Same values of excitation wavelength (532 nm) and power (100 µW) were used for both reference NV centre and MgO NPs [34]. The QE is then calculated by:(1)$$QE=QE_{NV}\;\times \;\frac{Int}{Int_{NV}}\;\times \;\frac{1-10^{-ANV}}{1-10^{-A}}\;\times \frac{n^{2}}{n_{NV}^{2}}$$
where *Int* is the area under the PL spectra, *A* is the absorption coefficient at 532 nm pump wavelength and *n* is the refractive index of the solvent. The formula gave a QE of 0.45 for MgO NPs.

### 3.4. Lifetime Measurements

The lifetime measurements were performed for both commercial and ball-milled MgO NPs. For the commercial NPs, a single lifetime was obtained. The data was recorded for several NPs and an average lifetime value of 1.37 ± 0.23 ns was attained. However, for the ball-milled MgO NPs, a different trend was observed: a slowly decaying lifetime component of 10.58 ± 1.20 ns and a fast decaying lifetime component of 2.06 ± 0.37 ns were observed.

Hence the commercial MgO NPs, that showed the presence of V^2+^ defects in the PL spectra, were found to have a single short lifetime, whereas the ball-milled MgO, that were fluorescent due to the Cr^3+^ defects, showed two lifetime components. These findings are consistent with the fact that the MgO NPs with the Cr^3+^ vacancy showed the presence of two distinct emission lines between 690–710 nm [22] The presence of two lifetimes for the ball-milled MgO NPs most likely corresponds to the two emission lines, R and N line, of Cr^3+^ defect as discussed in Section 3.3. As stated, the distinct features of those two defects, first spectral and now substantially different lifetimes, are extremely beneficial for multiparameter sensing.

### 3.5. Widefield Imaging of Cell Cultured NPs

The MgO NPs with emission in NIR wavelength range were employed for in vitro imaging in order to investigate their uptake, photostability and mobility. To check the stability of MgO NPs under cell culture conditions, DLS and zeta potential measurements were performed on 0.5 mg/mL of commercial MgO NPs in PBS and PBS + 10% FBS at 37 °C after 24 h, to simulate cell culture conditions. MgO NPs at the same concentration in water was used as a control. The DLS of MgO NPs in water shows four peaks ranging from ~2 nm to 4 µm (Appendix A), whereas in PBS alone, there in a monodispersed peak at approximately 1 µm. This suggests the agglomeration of the MgO NPs due to the ionic strength of the PBS [35]. The MgO NPs in PBS + 10% FBS, on the other hand, showed three peaks ranging from ~10 nm to 800 nm, and this suggests the proteins in the FBS prevented the agglomeration observed in PBS alone [35]. This occurs through the formation of a protein corona on the NPs, observed through the increase in size from ~2 nm to 10 nm, 43 nm to 53 nm and 400 nm to 802 nm between the NPs in water and in PBS+FBS [36]. This behavior was further confirmed by the zeta potential of the MgO NPs, which dropped from 13.8 ± 0.2 to −11.4 ± 0.3 mV in PBS, but remained at a higher value of −14.2 ± 0.2 in PBS+FBS. It should be noted that despite the change in size of the MgO NPs in PBS+FBS, this did not affect the uptake of the NPs in cells, as observed in Figure 6, Figure 7, Figure 8 and Figure 9.

Commercial and ball-milled MgO NPs were cultured in fibroblast and human keratinocytes cells, respectively, as shown in Figure 6 and Figure 7a,b. Cells were treated with 1 mg/mL of MgO NPs for ~16 h (overnight) before imaging. Since the cells were washed with PBS prior to imaging, the NPs observed in the images have been either taken up or are closely associated with the cell surface. The fluorescence images and time traces were recorded, within a detection window of 560–750 nm, for both types of NPs in 10–12 different cells.

The background subtracted fluorescence intensity for MgO NPs cultured inside the cells is shown in Figure 6c. The figure illustrates bright and stable emission from ball-milled NPs, within a range 35 k–0.5 M counts/s. The figure also shows an initial decay within the first few seconds attributed to partial photobleaching of cells [15]. This photobleaching is attributed to cell autofluorescence which diminishes in the first few seconds of exposure. A similar trend of in vitro fluorescence was seen with the commercial NPs inside human keratinocytes cells. Figure 7a shows the white light microscopic image of commercial MgO NPs cultured inside the cells while Figure 7b shows the red fluorescence of these NPs.

Hence both ball-milled and commercial MgO NPs exhibit bright red fluorescence both with and without cells (in widefield confocal scans). As discussed in the Introduction, their photostable NIR to infrared fluorescence meets the essential condition for in vivo imaging. It has also been demonstrated other studies that brightness and photostability can be further improved by encapsulation of NPs with a suitable biocompatible polymer [14,35,36].

To observe the mobility of ball-milled MgO NPs inside the fibroblast cells, the trajectories of the NPs in cells were recorded. After the video processing of the fluorescence images and tracking of the NP position, the diffusion coefficients (D) were calculated for six different MgO NPs, using the mean square displacements as reported in previous studies [14]. The data was extracted at a rate of seven frames per second resulting in a time step of 0.14 s. Diffusion coefficient (D) of NPs inside the cell membrane exhibited values ranging from a minimum of 0.0035 × 10^−2^ μm^2^/s to a maximum of 0.0250 × 10^−2^ μm^2^/s and a mean D of (1.20 ± 0.24) × 10^−4^ μm^2^/s.

According to the literature, very low diffusion coefficients ≤ 1.0 × 10^−2^ μm^2^/s are expected for NPs with diameters greater than 30 nm in cells [37]. Gold NPs of similar size range showed low mobilities of the order of 0.1 × 10 − 3 to 0.7 × 10^−2^ μm^2^/s [35]. Selenium NPs with 50 nm diameters are reported to give D values ranging from 0.1 × 10^−2^ to 2.2 × 10^−2^ μm^2^/s [14]. Nanodiamonds with diameter of 45 nm have been reported to provide an average diffusion coefficient of 0.3 × 10^−2^ μm^2^/s inside cells [37]. Hence the D values for MgO NPs exist in the range of values reported for other NPs in the size range of 30–50 nm. This is about three orders of magnitude smaller than the diffusion coefficient predicted for 30 nm spherical particles at infinite dilution according to the Stokes-Einstein equation (9.7 μm^2^/s, based on the viscosity of supplemented culture medium being 0.00078 Pa.s at 37 °C) [38]. A large reduction in the diffusivity of NPs in cells compared to their free solution behaviour is expected as the intracellular environment is a complex viscoelastic medium. NPs may reside in different intracellular structures with different rheological properties and could also experience active transport and/or specific interactions with particular cell components [38]. 

### 3.6. Toxicity Analysis of Commercial MgO NPs in Cancer Cells

In addition to human keratinocytes and fibroblasts, the fluorescence properties of NPs uptaken by cancer cells was investigated. Cancer cells were chosen for two reasons. First, it is known that cancer cells are more sensitive than normal cells to DNA damage response [39]. Second, on many occasions it has been reported that MgO nanoparticles can be selectively toxic towards cancer cells [20,21,22]. Additionally, MgO is used to relieve heartburn and indigestion and MgO nanoparticles have been shown to increase bone regeneration [39,40], hence they are considered biocompatible [3]. The cytotoxicity analysis was performed for the commercial MgO NPs and the results are reported in Appendix A. The MgO NPs showed dose dependent toxicity to prostate cancer (PC-3) cells from a low concentration of 62.5 μg/mL upwards, after 24 h of treatment. Hence, the previously reported claim that MgO NPs are toxic to cancer cells and can play a role in their inhibition was not supported by the present measurements [22]. Similar to our findings, it was reported that the genotoxic effect of MgO nanoparticles was not significant compared with control experiments when cells line with liver cancer epithelial cells were used [21]. However, it is hard to compare those findings as a wide range of different cell lines, nanoparticle properties and concentrations, exposure times, and colorimetric assays were used in those studies.

In order to record live images of prostate cancer cells cultured with MgO, the NPs were cultured in the cells for 12 h. Following uptake widefield fluorescence microscopy was performed. The images of Figure 8a,c show the white bright field microscopic image of the cancer cells, while Figure 8b,d show the fluorescent MgO NPs inside the cancer cells. Although the NPs were taken up by the cells, they did not cause any observable adverse effects. Note, that dead cells apparent in bright field images do not appear in confocal images because they did not uptake the MgO NPs. The fluorescence obtained for the NPs inside the cancer cells was brighter as compared to the healthy cells (Figure 6 and Figure 7), due to the higher uptake of NPs inside the cancer cells. The photostable fluorescence existed in a range of 50–150 k counts/s inside the cancer cells.

To compare functioning of MgO NPs to one of the standard staining experiments we used a combination of prostate cancer and healthy cells. E-Cadherin antibody was used as the primary antibody (antihuman rabbit antibody) and secondary antibody was goat antirabbit Alexafluor 594 to stain the human cancer cells. The normal cells were unstained. Human cancer cell line was PC-3 cells and the normal cells were mouse cell line J774. The stained cancer cells were compared with MgO cultured normal cells and the results are shown in Figure 9. The widefield fluorescence image of Figure 9b shows that antibody stained cancer cell (top left corner) showed lower brightness compared to the two MgO cultured normal cells (central region). Photobleaching occurred in antibody stained cells after few minutes, unless the samples are fixed and mounted using appropriate mounting medium. The MgO NPs, however, showed better photostability, in comparison, for the duration of the experiment.

## 4. Conclusions

We have investigated the photoluminescence properties of commercial and in-house ball-milled MgO NPs. Body temperature photostable bright emission was recorded and analysed for both samples. Emission spectra elucidate that Cr^3+^-related optical defects for ball-milled NPs and V^2+^-related optical defects for commercial NPs are responsible for this NIR emission. Lifetime measurements indicate two distinct life times for ball-milled NPs corresponding to two emission lines, which is extremely beneficial for multiparameter sensing. For the commercial NPs the emission line at 870 nm is an ideal candidate for in vivo imaging. Subsequently, these NPs were cultured in three types of cells and their fluorescence evaluated with wide field confocal imaging. Commercial and ball-milled MgO NPs were cultured in fibroblast and human keratinocyte cells, respectively. Our studies show that NIR emission from these centres is not only detectable, but also exhibits extremely photostable body temperature emission. Mobility of these NPs is comparable to other nanoparticle fluorescent markers reported in the literature. An analysis of the cytotoxicity in prostate cancer (PC-3) cells for the commercial MgO NPs showed dose dependent toxicity to cells only from 62.5 μg/mL upwards. It is expected that these initial, positive results will be the first step towards the widespread use of biodegradable MgO nanoparticles as fluorescent markers for short term bioimaging.

## Figures and Tables

**Figure 1 nanomaterials-09-01360-f001:**
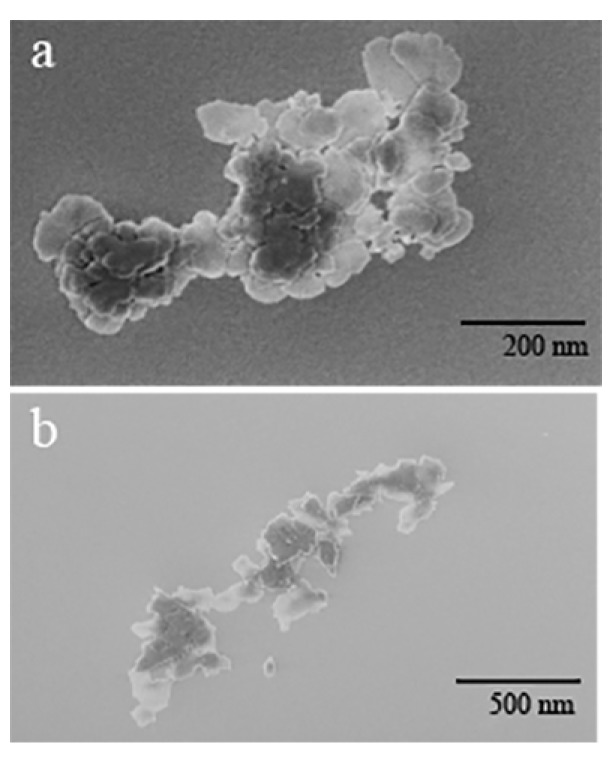
SEM images of (**a**) commercial MgO nanoparticles and (**b**) 2000 rpm pellet of ball-milled MgO nanoparticles. The samples were deposited on strong carbon grids and the electron beam was accelerated with 500 kV.

**Figure 2 nanomaterials-09-01360-f002:**
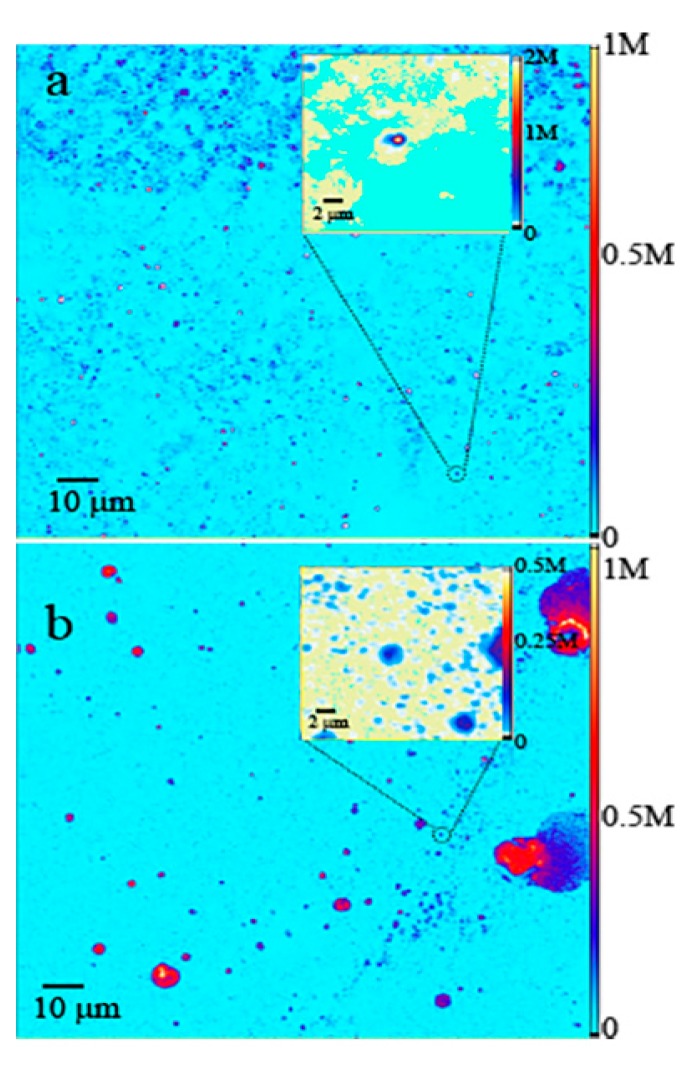
(**a**) Two-dimensional confocal fluorescence maps (100 × 100 μm^2^) for (**a**) commercial and (**b**) ball-milled MgO NPs, excited with 200 µW power. The insets show fine scans of 10 × 10 μm^2^ dimensions. The fluorescence intensity detected by the SPAD is expressed in counts per second (counts/s).

**Figure 3 nanomaterials-09-01360-f003:**
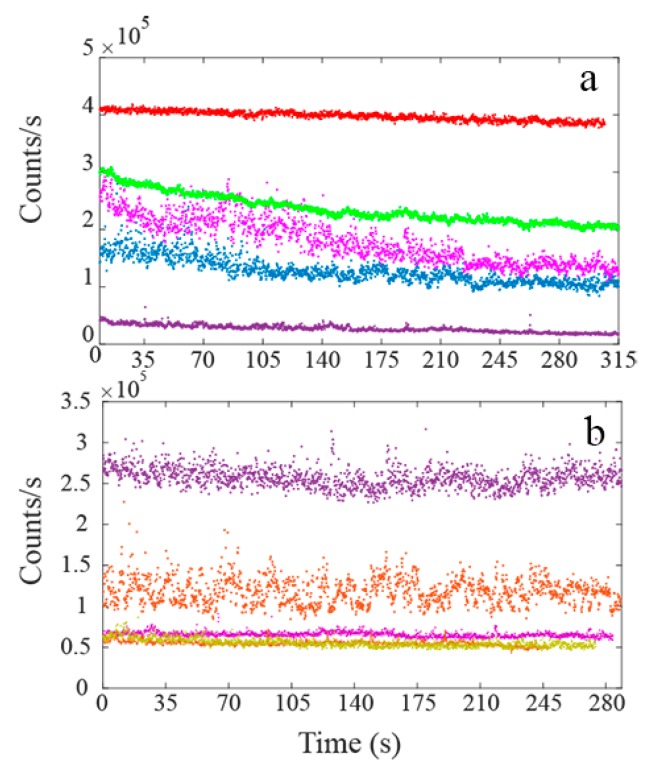
Emission traces (dotted) for individual MgO nanoparticles from (**a**) commercial and (**b**) ball-milled samples, excited with 532 nm pump at a power of 200 μW. The average background counts of 10 k are represented with the solid black lines in the plots.

**Figure 4 nanomaterials-09-01360-f004:**
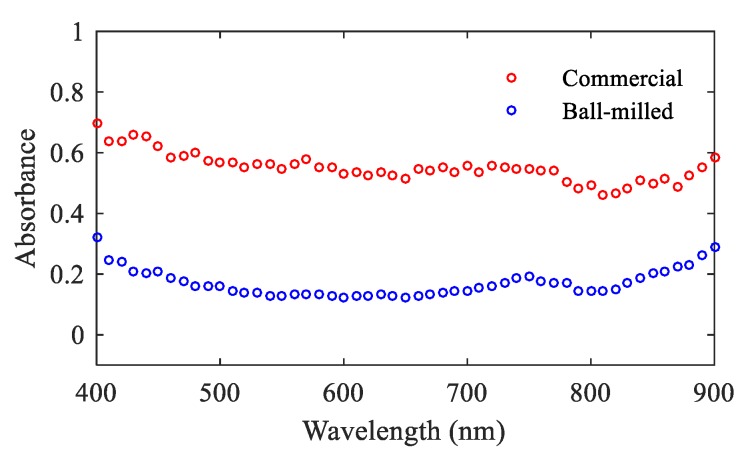
Absorbance spectra for ball-milled (blue) and commercial (red) MgO NPs. The y-axis shows absorbance in normalized units. The dashed line indicates the 532 nm wavelength used to excite the NPs for fluorescence. A concentration of 0.5 mg/mL was used both for the commercial and ball-milled MgO NPs.

**Figure 5 nanomaterials-09-01360-f005:**
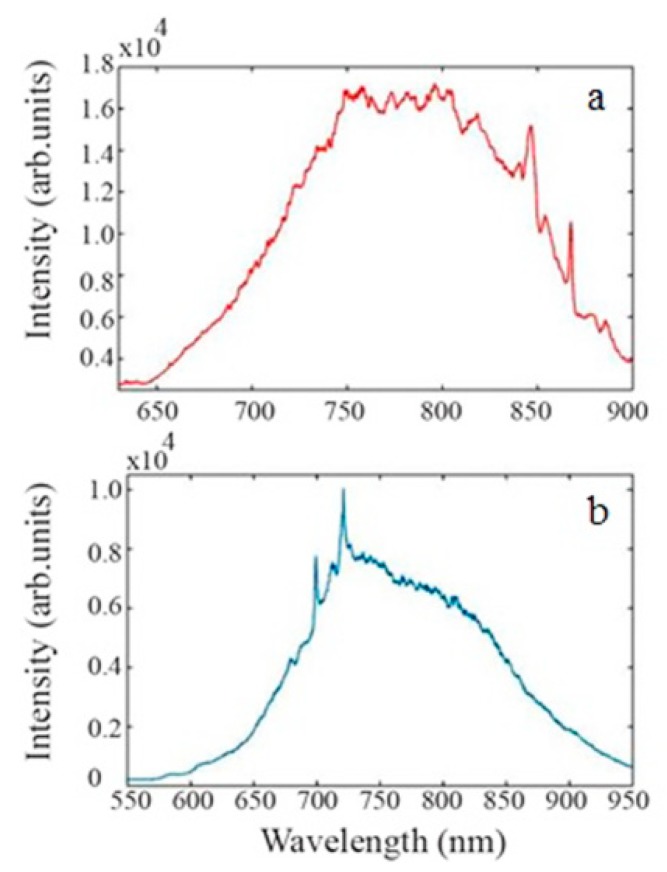
Photoluminescence spectra under 532 nm excitation of (**a**) commercial and (**b**) ball-milled and MgO nanoparticles.

**Figure 6 nanomaterials-09-01360-f006:**
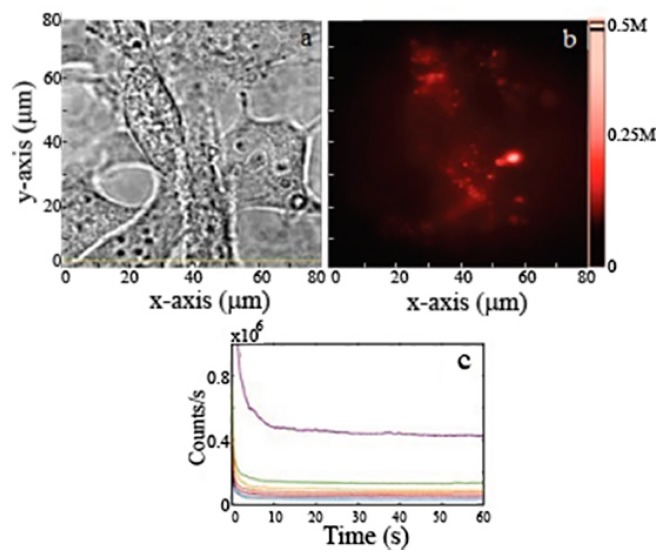
(**a**) Brightfield and (**b**) widefield fluorescence images, fibroblast cells cultured with ball-milled MgO NPs. (**c**) Emission counts recorded as a function of time for NPs cultured in different cells. Excitation power was 300 mW at the back of the objective distributed over a widefield scan area of 80 × 80 µm^2^. This provides an average power of 45 µW/µm^2^ on the sample’s surface.

**Figure 7 nanomaterials-09-01360-f007:**
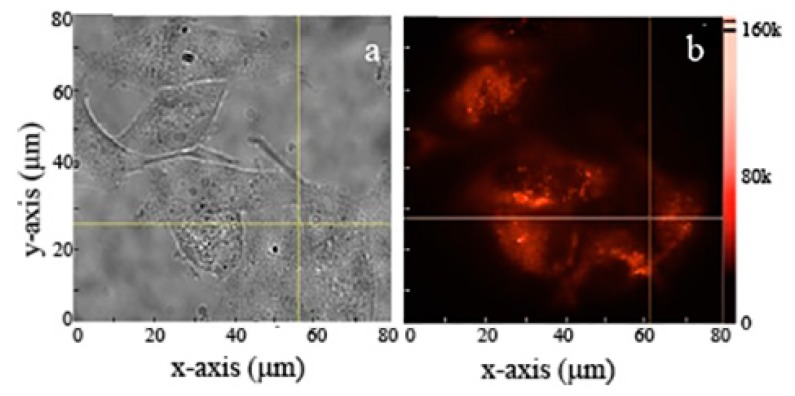
(**a**) Brightfield and (**b**) widefield fluorescence image of human keratinocytes cells cultured with commercial MgO NPs. Excitation power was 300 mW at the back of the objective distributed over a widefield scan area of 80 × 80 µm^2^. This provides an average power of 45 µW/µm^2^ on the sample’s surface.

**Figure 8 nanomaterials-09-01360-f008:**
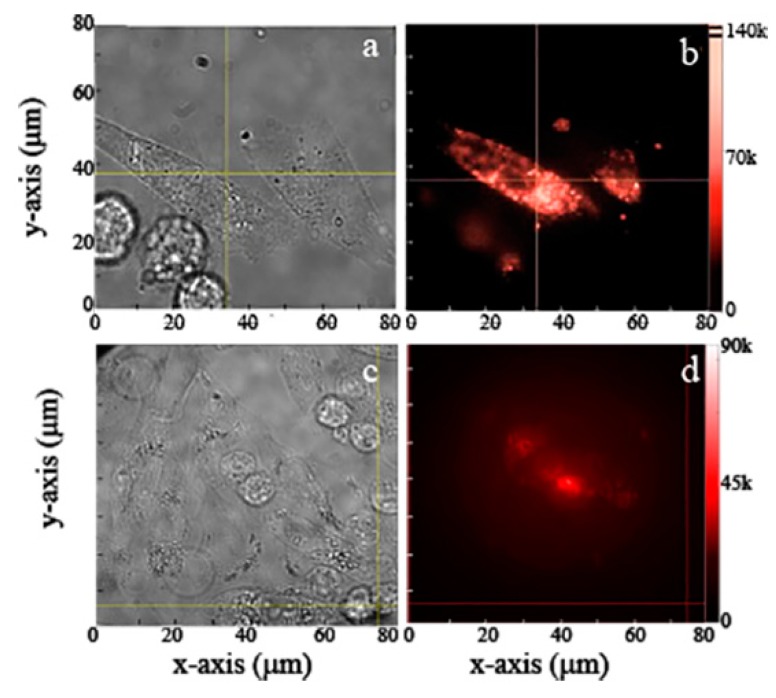
Prostate cancer cells cultured with (**a**,**b**) commercial and (**c**,**d**) ball-milled MgO NPs. (**a**,**c**) Bright field and (**b**,**d**) widefield fluorescence images of selected areas on the sample. Excitation power was 300 mW at the back of the objective distributed over a widefield scan area of 80 × 80 µm^2^. This provides an average power of 45 µW/µm^2^ on the sample’s surface.

**Figure 9 nanomaterials-09-01360-f009:**
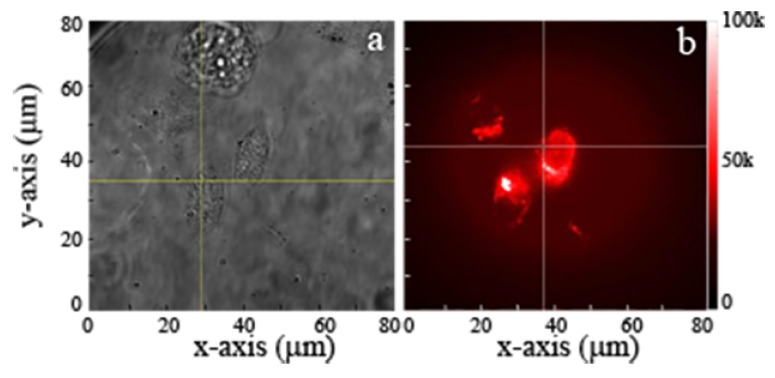
Prostate cancer cells and healthy cells (**a**) bright field and (**b**) widefield fluorescence of cells cultured with commercial MgO NPs; Cancer cells were stained with Alexafluor 594 without MgO NPs.

**Table 1 nanomaterials-09-01360-t001:** MgO particle sizes for a range of centrifuging speeds. The polydispersity index (PDI) for each measurement is also shown in the table.

**Centrifuge Speed (rpm)**	1000	2000	3000
**Pellet Size (nm)**	1640 ± 630	1060 ± 480	760 ± 60
**PDI**	0.25 ± 0.01	0.56 ± 0.04	0.46 ± 0.02
**Supernatant Size (nm)**	300 ± 6	180 ± 4	100 ± 2
**PDI**	4.0 × 10^−4^	4.8 × 10^−4^	4.0 × 10^−4^

**Table 2 nanomaterials-09-01360-t002:** Percentage present of impurities within the MgO ball-milled sample measured using ICP-MS.

**Impurity**	Si	Cr	Ni	V	Al	Fe	Ca
**Percentage**	<0.1	<0.01	0.02	<0.01	0.01	<0.05	<0.05

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
