# Peer review of "Biocompatible and Biodegradable Magnesium Oxide Nanoparticles with In Vitro Photostable Near-Infrared Emission: Short-Term Fluorescent Markers"

_nanomaterials, 2019, doi:10.3390/nano9101360_

Round 1
Reviewer 1 Report
The authors report preliminary findings on the use of magnesium oxide nanoparticles from both a commercial source and ball-mill processed materials as potential bioimaging probes. In this materials characterization and in vitro based study, the authors seek to establish a foundation for the use of these nanomaterials for biotechnological and medical applications. Although this report presents a coherent body of work on this materials, the reviewer is concerned about some aspects of the study and requests clarification. The following comments/recommendations are made to improve the quality of the study for publication in this journal:
Overall, the photophysical and materials characterization are well presented to support the goals of the study. Figure 4, the concentration used to obtain the absorbance spectra seems high. Please provide spectra at a lower concentration. The materials evaluated without surface coating, which is generally necessary for biological applications. Please provide some characterization of the surface of these nanoparticles, such as zeta potential. Biological assays can be improved to support the claim of biocompatibility Although the authors provide some justification for using cancer cells for viability studies it is still unclear why data on toxicity toward keratinocytes and fibroblasts are not included. Figure 9, missing panels c & d.
Author Response
Referee: 1
Comment 1.1:
Figure 4, the concentration used to obtain the absorbance spectra seems high. Please provide spectra at a lower concentration.
Response 1.1:
As suggested by the reviewer we have measured and provided the absorbance spectra at a lower concentration (new Figure 4). A concentration of 0.5 mg/mL was used both for the commercial and ball-milled MgO NPs.
Comment 1.2:
Please provide some characterization of the surface of these nanoparticles, such as zeta potential.
Response 1.2:
Zeta potential was provided on page 4, 2nd column, 2nd paragraph:
“An average zeta potential value of 13.8±0.2 for the commercial MgO and 15.83±4.27 mV for the ball-milled MgO NPs is observed in water, which is expected to contribute to partial stabilization of the NPs in the solution by maintaining the electrostatic repulsion between the particles.” However, we did additional measurements of zeta potential in PBS. On page 6, 2nd column:
“The zeta potential in PBS dropped to -11.4±0.3 mV and a large size distribution averaged around 1µm was observed within 1h and 24 h of incubation.” Moreover, additional zeta potential and DLS measurements are made in water, PBS and culture medium, as mentioned in methods Section 27 and results Section 3.5 as follows.
2.7 Stability in cell culture media and cytotoxicity analysis in cancer cells
To study the stability of the MgO NPs under cell culture conditions, the MgO NPs were incubated in PBS and PBS containing 10 % foetal bovine serum at 37 °C. Dynamic light scattering (DLS) and zeta potential measurements were conducted after 24 hours incubation, on a Malvern Zetasizer. The measurements were performed at 25ºC and each sample and 6 scans were performed for each sample.
3.5 Wide-field imaging of cell cultured NPs
The MgO NPs with emission in NIR wavelength range were employed for in vitro imaging in order to investigate their uptake, photostability and mobility. To check the stability of MgO NPs under cell culture conditions, DLS and zeta potential measurements were performed on 0.5 mg/mL of commercial MgO NPs in PBS and PBS+10% FBS at 37 °C after 24 hours, to simulate cell culture conditions. MgO NPs at the same concentration in water was used as a control. The DLS of MgO NPs in water shows four peaks ranging from ~ 2 nm to 4 µm (Supplementary Information Figure S5), whereas in PBS alone, there in a monodispersed peak at approximately 1 µm. This suggests the agglomeration of the MgO NPs due to the ionic strength of the PBS. [1] The MgO NPs in PBS+10% FBS, on the other hand, showed three peaks ranging from ~ 10 nm to 800 nm, and this suggests the proteins in the FBS prevented the agglomeration observed in PBS alone. [1] This occurs through the formation of a protein corona on the NPs, observed through the increase in size from ~2 nm to 10 nm, 43 nm to 53 nm and 400 nm to 802 nm between the NPs in water and in PBS+FBS. [2] This behavior was further confirmed by the zeta potential of the MgO NPs, which dropped from 13.8±0.2 to -11.4±0.3 mV in PBS, but remained at a higher value of -14.2±0.2 in PBS+FBS. It should be noted that despite the change in size of the MgO NPs in PBS+FBS, this did not affect the uptake of the NPs in cells, as observed in Figures 7-9.
Comment 1.3:
Although the authors provide some justification for using cancer cells for viability studies it is still unclear why data on toxicity toward keratinocytes and fibroblasts are not included.
Response 1.3:
There are two main reasons why we have provided cytotoxicity analysis in cancer cells only. As pointed out in Section 3.6, 2nd sentence: “Cancer cells were chosen for two reasons. First, it is known that cancer cells are more sensitive than normal cells to DNA damage response.38 Second, on many occasions it has been reported that MgO nanoparticles can be selectively toxic towards cancer cells.20-22. Additionally, MgO is used to relieve heartburn and indigestion and MgO nanoparticles have been shown to increase bone regeneration hence they are considered biocompatible. [3]
Comment 1.4:
Figure 9, missing panels c & d.
Response 1.4:
As suggested by the reviewers Figure 9 has been fixed.
References
Du, S., et al., Aggregation and adhesion of gold nanoparticles in phosphate buffered saline. Journal of Nanoparticle Research, 2012. 14(3): p. 758. Vidic, J., et al., Effects of Water and Cell Culture Media on the Physicochemical Properties of ZnMgO Nanoparticles and Their Toxicity toward Mammalian Cells. Langmuir, 2014. 30(38): p. 11366-11374. Roh, H.-S., et al., Addition of MgO nanoparticles and plasma surface treatment of three-dimensional printed polycaprolactone/hydroxyapatite scaffolds for improving bone regeneration. Materials Science and Engineering: C, 2017. 74: p. 525-535. Schwaiger, R., et al., Hydration of magnesia cubes: a helium ion microscopy study. Beilstein journal of nanotechnology, 2016. 7: p. 302-309. Khalid, A., et al., Fluorescent Nanodiamond Silk Fibroin Spheres: Advanced Nanoscale Bioimaging Tool. ACS Biomaterials Science & Engineering, 2015. 1(11): p. 1104-1113. Khalid, A., et al., Intrinsic fluorescence of selenium nanoparticles for cellular imaging applications. Nanoscale, 2016. 8(6): p. 3376-85. Ramanavicius, A., et al., Stabilization of (CdSe)ZnS quantum dots with polypyrrole formed by UV/VIS irradiation initiated polymerization. J Nanosci Nanotechnol, 2009. 9(3): p. 1909-15. Inam, F.A., et al., Emission and Nonradiative Decay of Nanodiamond NV Centers in a Low Refractive Index Environment. ACS Nano, 2013. 7(5): p. 3833-3843. Brouwer Albert, M., Standards for Photoluminescence Quantum Yield Measurements in Solution.

Reviewer 2 Report
Reviewer report on Manuscript Draft entitled ‘Biocompatible and biodegradable magnesium oxide nanoparticles with in vitro photostable near-infrared emission: short-term fluorescent
markers’
In this paper, authors report the photoluminescent properties of magnesium oxide (MgO) nanoparticles (NPs) that meet all these criteria. These NPs have been successfully integrated into cultured cells and photostable bright in vitro emission from NPs was recorded and analyzed. Authors expect that numerous biotechnological and medical applications will emerge as this nanomaterial satisfies all criteria for short-term in vivo imaging. The manuscript is well written and well addressed. Presented discussions are interesting from technological and analytical points of view. The research is in scope of the journal. Therefore, the manuscript can be published after some minor improvements and corrections:
The legend ‘Figure 5: Optical spectra under 532 nm excitation of (a) commercial and (b) ball-milled and MgO nanoparticles.’ Should be corrected is not clear what authors mean under ‘Optical spectra’ – because there are many very different types of optical spectra, the most probably it is ‘Photoluminescence spectra’.
Authors declare that MgO nanoparticles are quickly degrading (‘It is expected that these initial, positive results will be the first step towards the widespread use of biodegradable MgO nanoparticles as fluorescent markers for short term bioimaging.’). It will be nice if authors will present how the Photoluminescence spectra is changing `within time, there is significant blue shift in Photoluminescence maximum or not?
In order to get more stabile Photoluminescent-MgO-NPs strategies for the stabilization of here reported MgO nanoparticles (Stabilization of (CdSe)ZnS quantum dots with polypyrrole formed by UV/VIS irradiation initiated polymerisation. Journal of Nanosciense and Nanotechnology 2009, 9, 1909–1915.) that were demonstrated on other Photoluminescent-NPs and can be based on encapsulation of NPs within biocompatible conducting polymers should be discussed as one possible future trend in the development of Photoluminescent-MgO-NPs.
Author Response
Referee: 2
Comment 2.1:
The legend ‘Figure 5: Optical spectra under 532 nm excitation of (a) commercial and (b) ball-milled and MgO nanoparticles.’ Should be corrected is not clear what authors mean under ‘Optical spectra’ – because there are many very different types of optical spectra, the most probably it is ‘Photoluminescence spectra’.
Response 2.1:
The reviewer is correct that optical spectra has too broad meaning. Now the caption of Fig 5 is: “Photoluminescence spectra under 532 nm excitation of (a) commercial and (b) ball-milled and MgO nanoparticles.
Comment 2.2:
Authors declare that MgO nanoparticles are quickly degrading (‘It is expected that these initial, positive results will be the first step towards the widespread use of biodegradable MgO nanoparticles as fluorescent markers for short term bioimaging.’). It will be nice if authors will present how the Photoluminescence spectra is changing `within time, there is significant blue shift in Photoluminescence maximum or not?
Response 2.2:
Even though we have not observed blue shift in PL maximum, we are aware that it has been reported in literature. As suggested by the reviewer we add this discussion to Section 3.3 (emission spectra), page 6, column 2.
Some reports underline the difficulties of maintaining desired nanomaterial properties during handling and processing due to their dynamic nature. [4] In particular, blue shift in PL maximum was reported for l MgO cubes (size dependant) due to the formation of Mg(OH)2 shells that act as diffusion barriers for MgO dissolution.
Comment 2.3:
In order to get more stabile Photoluminescent-MgO-NPs strategies for the stabilization of here reported MgO nanoparticles (Stabilization of (CdSe)ZnS quantum dots with polypyrrole formed by UV/VIS irradiation initiated polymerisation. Journal of Nanosciense and Nanotechnology 2009, 9, 1909–1915.) that were demonstrated on other Photoluminescent-NPs and can be based on encapsulation of NPs within biocompatible conducting polymers should be discussed as one possible future trend in the development of Photoluminescent-MgO-NPs.
Response 2.3:
According to reviewer’s suggestion, we are currently working on biopolymer coated MgO NPs for the development of more stable and brighter NPs for biomarking applications.
The reviewer’s suggestion has been incorporated in Section 3.3 (Emission spectra), Page 6, last paragraph.
It has also been demonstrated in earlier studies that brightness and photostability can be further improved by encapsulation of NPs with a suitable biocompatible polymer. [5-7]
References
Du, S., et al., Aggregation and adhesion of gold nanoparticles in phosphate buffered saline. Journal of Nanoparticle Research, 2012. 14(3): p. 758. Vidic, J., et al., Effects of Water and Cell Culture Media on the Physicochemical Properties of ZnMgO Nanoparticles and Their Toxicity toward Mammalian Cells. Langmuir, 2014. 30(38): p. 11366-11374. Roh, H.-S., et al., Addition of MgO nanoparticles and plasma surface treatment of three-dimensional printed polycaprolactone/hydroxyapatite scaffolds for improving bone regeneration. Materials Science and Engineering: C, 2017. 74: p. 525-535. Schwaiger, R., et al., Hydration of magnesia cubes: a helium ion microscopy study. Beilstein journal of nanotechnology, 2016. 7: p. 302-309. Khalid, A., et al., Fluorescent Nanodiamond Silk Fibroin Spheres: Advanced Nanoscale Bioimaging Tool. ACS Biomaterials Science & Engineering, 2015. 1(11): p. 1104-1113. Khalid, A., et al., Intrinsic fluorescence of selenium nanoparticles for cellular imaging applications. Nanoscale, 2016. 8(6): p. 3376-85. Ramanavicius, A., et al., Stabilization of (CdSe)ZnS quantum dots with polypyrrole formed by UV/VIS irradiation initiated polymerization. J Nanosci Nanotechnol, 2009. 9(3): p. 1909-15. Inam, F.A., et al., Emission and Nonradiative Decay of Nanodiamond NV Centers in a Low Refractive Index Environment. ACS Nano, 2013. 7(5): p. 3833-3843. Brouwer Albert, M., Standards for Photoluminescence Quantum Yield Measurements in Solution.

Reviewer 3 Report
In this article, the authors examine the fluorescence properties of MgO nanoparticles obtained from commercial sources or by ball milling. This material is potentially interesting thanks to its biodegradability and biocompatibility. They report that MgO nanoparticles exhibit fluorescence when excited at 532 nm. They then show that MgO nanoparticles retain their fluorescence after uptake by cells in culture. The work is novel and brings new information but remains very preliminary. It is unlikely that these particles will find an application in the present state (no morphology control, no functionalization).
The morphology and size distribution of the obtained nanoparticles are not well controlled as shown in the TEM images.
The optical characterization is interesting but incomplete.
- What is the typical extinction coefficient of the nanoparticles at 532 nm? This could be probably inferred from solution phase absorption measurements at known mass concentrations. Extinction coefficients of particles could then be estimated from the average particle size and density.
- What is the quantum yield of these nanoparticles?
Are these nanoparticles colloidally stable in the culture medium?
In the cytotoxicity assay (Figure S3, not S2), the concentration at which nanoparticles start to induce toxicity seems to be much less than 250 µg/mL, but starting from 15-30 µg/mL.
I could not find the concentration of MgO used in the cell imaging experiments. If this information is absent, the authors should add it in the experimental section or in the text.
Figure 9c,d panels are missing, which prevents evaluation of the data.
Author Response
Referee: 3
Comment 3.1:
What is the typical extinction coefficient of the nanoparticles at 532 nm? This could be probably inferred from solution phase absorption measurements at known mass concentrations. Extinction coefficients of particles could then be estimated from the average particle size and density.
Response 3.1:
As suggested by the reviewer we estimated the extinction coefficient to be in the range k=0.02 for commercial to k=0.03 for ball-milled MgO NPs at the pump wavelength, l=532 nm.
Comment 3.2:
- What is the quantum yield of these nanoparticles?
Response 3.2:
The yield of MgO NPs was measured by comparing their photoluminescence spectra to those of nitrogen vacancy (NV) centre in nanodiamonds. The NV centre in nanodiamonds [8] have a known quantum yield of 0.7. [8] Same values of excitation wavelength (532 nm) and power (100 µW) were used for both reference NV centre and MgO NPs. [9]The quantum yield is then calculated by
Where QE is the quantum yield or quantum efficiency, Int is the area under the PL spectra, A is the absorption coefficient at 532 nm pump wavelength and n is the refractive index of the solvent.
The formula gave a QE of 0.45 for MgO NPs.
The reviewer’s suggestion is incorporated to the Section 3.3 (Emission spectra) of the manuscript on page 6.
Comment 3.3:
Are these nanoparticles colloidally stable in the culture medium?
Response 3.3:
As suggested by the reviewer, the MgO NPs were tested in PBS and PBS+10% FBS and the results have been added to the manuscript as highlighted in Section 3.5 as well as to the Figure S4 of the supporting information file. The additions are also listed above in Response 1.3 to reviewer 1. The following text is added to Section S4 of the supporting information.
Hence the MgO NPs are considered stable in PBS+10% FBS, under cell culture conditions, in comparison to the MgO NPs in PBS alone. Extensive aggregation was observed in PBS, however in the presence of 10% FBS, protein corona formation was seen after 24 h. This is a commonly observed through the increase in hydrodynamic size of the NPs, as we similarly observed with the MgO NPs.
Comment 3.4:
In the cytotoxicity assay (Figure S3, not S2), the concentration at which nanoparticles start to induce toxicity seems to be much less than 250 µg/mL, but starting from 15-30 µg/mL.
Response 3.4:
The reviewer is correct in saying to induced toxicity seems to be much less than 250 µg/mL, it seems that we accidently put the wrong value, but induced toxicity starts from 62.5 ug/mL. However, it is not likely to be 15-30 ug/mL since the error bars show no significant difference between them and the untreated (0 ug/mL) cells. We corrected the number in our supplementary material.
Comment 3.5:
I could not find the concentration of MgO used in the cell imaging experiments. If this information is absent, the authors should add it in the experimental section or in the text.
Response 3.5:
Cells were treated with 1mg/mL of MgO NPs for ~16 hours (overnight) before imaging. This information is added to Section 3.5, page 7 of the revised manuscript.
Comment 3.6:
Figure 9c,d panels are missing, which prevents evaluation of the data.
Response 3.6:
Fig 9 has been fixed.
References
Du, S., et al., Aggregation and adhesion of gold nanoparticles in phosphate buffered saline. Journal of Nanoparticle Research, 2012. 14(3): p. 758. Vidic, J., et al., Effects of Water and Cell Culture Media on the Physicochemical Properties of ZnMgO Nanoparticles and Their Toxicity toward Mammalian Cells. Langmuir, 2014. 30(38): p. 11366-11374. Roh, H.-S., et al., Addition of MgO nanoparticles and plasma surface treatment of three-dimensional printed polycaprolactone/hydroxyapatite scaffolds for improving bone regeneration. Materials Science and Engineering: C, 2017. 74: p. 525-535. Schwaiger, R., et al., Hydration of magnesia cubes: a helium ion microscopy study. Beilstein journal of nanotechnology, 2016. 7: p. 302-309. Khalid, A., et al., Fluorescent Nanodiamond Silk Fibroin Spheres: Advanced Nanoscale Bioimaging Tool. ACS Biomaterials Science & Engineering, 2015. 1(11): p. 1104-1113. Khalid, A., et al., Intrinsic fluorescence of selenium nanoparticles for cellular imaging applications. Nanoscale, 2016. 8(6): p. 3376-85. Ramanavicius, A., et al., Stabilization of (CdSe)ZnS quantum dots with polypyrrole formed by UV/VIS irradiation initiated polymerization. J Nanosci Nanotechnol, 2009. 9(3): p. 1909-15. Inam, F.A., et al., Emission and Nonradiative Decay of Nanodiamond NV Centers in a Low Refractive Index Environment. ACS Nano, 2013. 7(5): p. 3833-3843. Brouwer Albert, M., Standards for Photoluminescence Quantum Yield Measurements in Solution.

Round 2
Reviewer 1 Report
The authors have addressed this reviewers concerns.